# Physicians’ and Nurses’ Knowledge in Palliative Care: Multidimensional Regression Models

**DOI:** 10.3390/ijerph18095031

**Published:** 2021-05-10

**Authors:** Jaime Martín-Martín, Mónica López-García, María Dolores Medina-Abellán, Cristina María Beltrán-Aroca, Stella Martín-de-las-Heras, Leticia Rubio, María Dolores Pérez-Cárceles

**Affiliations:** 1Department of Human Anatomy and Legal Medicine, Instituto de Investigación Biomédica de Málaga (IBIMA), School of Medicine, University of Málaga, 29071 Málaga, Spain; jaimemartinmartin@uma.es (J.M.-M.); monicalopez@cudeca.org (M.L.-G.); smdelasheras@uma.es (S.M.-d.-l.-H.); 2Fundación CUDECA, Benalmádena, 29631 Málaga, Spain; 3Health Center in Espinardo, Health Service, 30100 Murcia, Spain; medinaabellan@hotmail.com; 4Sección de Medicina Legal y Forense, Facultad de Medicina y Enfermería, Universidad de Córdoba, 14004 Córdoba, Spain; cristinabeltran@uco.es; 5Department of Legal and Forensic Medicine, School of Medicine, Biomedical Research Institute (IMIB-Arrixaca), University of Murcia, 30100 Murcia, Spain; mdperez@um.es

**Keywords:** knowledge, education, Palliative Care Knowledge Test Spanish Version (PCKT-SV), physician, nurse, palliative care, legal medicine

## Abstract

The increase in life expectancy has led to a growth in the number of people in need of palliative care. Health professionals must possess appropriate knowledge and skills. This study aimed to assess knowledge in palliative care through the Palliative Care Knowledge Test Spanish Version (PCKT-SV)^®^. A cross-sectional analytical study was conducted in 40 primary care health services. A total of 600 PCKT-SV questionnaires were distributed among health professionals; 561 of them (226 nurses and 335 physicians) were properly filled up. Sociodemographic information, education, and work experience were also recorded. A total of 34.41% of the nurses and 67.40% of the physicians showed good or excellent knowledge of palliative care. Physicians’ scores for pain, dyspnea, and psychiatric disorders were higher than those of the nurses. Nurses scored significantly better in philosophy. Professionals with continuous training in palliative care showed a higher level of knowledge. Age and work experience of physicians and undergraduate training in nurses had significant weight in knowledge. Developing continuous training and enhancing undergraduate training in palliative care will lead to improved patient care at the end of life.

## 1. Introduction

Knowledge of palliative care is essential in ensuring the delivery of professional care. Globally, more than 40 million people annually need palliative care. The number of people over the age of 60 in the world population will almost double by 2050 [1,2]. A similar phenomenon can also be seen in Europe, where the aging population has been increasing for several decades, and the number of people aged over 65 is expected to increase further [3]. Moreover, the increase in chronic diseases, cancer, respiratory and heart diseases, and stroke, among others, will also require palliative care [4]. Therefore, more palliative care support will be required for some illnesses and clinical syndromes, such as multimorbidity, chronic progressive illnesses, or advanced-stage diseases [5,6,7]. In addition, at this time, the COVID-19 pandemic has led to increased needs for palliative care [8]. Health and social care services will need to provide appropriate services for demographically changing populations and must ensure the availability of palliative care to help people with severe COVID-19. This will require greater knowledge in palliative care to obtain quality care and assistance for these patients.

The provision of high-quality palliative care to the growing number of older people living is a challenge in European countries, and it requires that staff possess appropriate knowledge and skills [9]. However, most studies suggest that recently graduated physicians and nurses are poorly prepared for end-of-life care [10,11,12,13]. Furthermore, students and health professionals identify a lack of education in palliative care, both in theory and in practice, and do not feel capable of managing patients who need this care [12,14,15].

Effective measurement of the knowledge of physicians is an essential component of the assessment of educational skills in palliative care. However, most studies have been conducted to assess the knowledge of palliative care nurses exclusively [13,16,17,18]. In this sense, Palliative Care Knowledge Test (PCKT) assesses the knowledge of both physicians and nurses [19]. The PCKT is a self-administered questionnaire made up of 20 multiple-choice items based on five domains, including “philosophy”, “pain”, dyspnea”, “psychiatric disorders”, and “gastrointestinal disorders”. The original PCKT was used to estimate the effects of the palliative care training program (PEACE) design for 57,664 physicians in Japan [20]. The PCKT has been translated into other languages [21,22,23,24] and used in various studies among community health care providers [20,25].

In a previous study, specifically in April 2020, we validated and registered a Spanish version of the PCKT (Palliative Care Knowledge Test Spanish Version, PCKT-SV^®^) in a population of 561 nursing and medicine professionals from different public health care centers, obtaining an internal consistency of the questionnaire of 0.741 and a content validity rate of 0.87 [23]. The current research goes further and aims to determine the palliative care knowledge of physicians and nurses using the PCKT-SV^®^.

## 2. Materials and Methods

### 2.1. Design

A cross-sectional study and analytical online survey design were carried out to assess the knowledge of palliative care based on the PCKT-SV^®^ of nurses and physicians. Sociodemographic variables, education, and work experience were also recorded.

### 2.2. Setting

The study was developed in 40 Primary Care Health Services in the Murcia Region (Southeast Spain) in 2016. Six hundred PCKT-SV questionnaires were distributed among health professionals (physicians and nurses); 588 of them responded to the surveys and, after a revision, we verified that 561 were properly filled up.

### 2.3. Instrument

The questionnaire consisted of two parts. The first section included demographic information: age, sex, training (nurse or physician), years of work experience, training received in palliative care in the degree, continuing training received in the last 5 years, and work experience in palliative care. The second part was the PCKT-SV^®^ [23], adapted and validated from the Palliative Care Knowledge Test (PCKT) [19,26]. The PCKT-SV is made up of 20 true or false questions structured in five dimensions (philosophy, pain, respiratory disorders, neuropsychiatric disorders, and gastrointestinal disorders). This test was chosen because it allows us to assess not only the knowledge in physicians and nurses jointly, but also because it can be used to improve educational programs [19].

### 2.4. Procedure

The PCKT-SV questionnaires were distributed by email, with attached files to record data on their profession, specialty, sociodemographic information, level of training in palliative care (undergraduate, postgraduate, and continuous training), and work experience in palliative care. Undergraduate training was considered if the participants had received theoretical or practical training on the PCKT-SV dimensions during their university studies. Postgraduate and continuous training was considered if the health professionals had participated in palliative care courses after their degrees. Work experience in palliative care was defined as the clinical assistance to any patient with palliative care requirements.

An identification number was given to each questionnaire received. Information about the purpose of the research, confirmation of the anonymity of each participant, informed consent, and an informative letter were sent in the same email.

### 2.5. Statistical Analysis

Descriptive analysis (mean and standard deviation) was performed for the continuous variables (age and years of practice). Analysis of categorical variables was carried out by frequency and proportion. The responses to the PCKT-SV questionnaire were compared between the groups of participating nurses and physicians. The comparison between groups was performed by Student’s t-test after checking the normality assumptions, the Kolmogorov–Smirnov test, and the homogeneity of variances with the Levene test. A multiple linear regression model was applied to determine the significant effect of the dependency variables on the level of knowledge in palliative care in general, and on its dimensions. SPSS version 23 (IBM^®^ SPSS Statistics^®^, IBM Software Group, Chicago, IL, USA) was used for statistical analyses, and *p* < 0.05 was considered significant.

### 2.6. Ethical Statements

Written informed consent was obtained from each participant after a complete description of the study. The study was approved by the Human Research Ethics Committee of the University of Murcia (Approval number: 2388/2018) in accordance with the “Ethical Principles for Medical Research Involving Human Subjects” adopted in the Declaration of Helsinki by the World Medical Association (64th WMA General Assembly, Fortaleza, Brazil, October 2013) and Spanish data protection (*Ley Orgánica 3/2018 de Protección de Datos Digitales*).

## 3. Results

The study was conducted with 561 health professionals, of which 38.5% were men and 61.5% were women, aged between 25 and 73 years, with an average age of 41.7 years (SD ± 11.3). A total of 59.7% of the participants were physicians, and 40.3% were nurses, with a total average of 15.2 years of working experience in their profession (min–max: 1–42, SD ± 10.1). The majority of men were physicians (47.8%). The majority of nurses were women (75.2%). Undergraduate palliative training was higher in nurses (44.2%) than in physicians (19.4%). Most of the physicians and nurses had no post-graduate (59.1% and 82.3%, respectively) or work experience in palliative care (70.1% and 71.2%, respectively) (Table 1).

PCKT-SV scores showed adequate/good general knowledge in palliative care (10.7 ± 3.2). The best-scored dimension was philosophy, and the dimension with the lowest score was pain. Statement 6 (*The effect of opioids should decrease when pentazocine or buprenorphine hydrochloride is used together after opioids are used*) obtained the lowest percentage of correct answers (18.5%), while statement 15 (*Some dying patients will require continuous sedation to alleviate suffering*) was the one with the highest success rate (92.5%).

No significant differences were shown between men (11.24 ± 3.58) and women (10.48 ± 3.16) in palliative care global knowledge. Comparing dimensions, the pain and dyspnea scores of men were significantly higher than those of women.

According to the type of profession (Table 2), the level of general knowledge of physicians (11.85 ± 3.2) was significantly higher than nurses’ knowledge (9.16 ± 2.88). Regarding dimensions, physicians’ scores for pain, dyspnea, and psychiatric disorders were significantly higher than those of the nurses. However, nurses scored significantly better in philosophy.

The participants’ global knowledge in palliative care was structured into four groups according to the number of correct answers: group 1 (excellent) 100–76%, group 2 (good) 75–51%, group 3 (poor) 50–26%, and group 4 (bad) 25–0% based on Ioshimoto et al. 2020 [21].

Figure 1 depicts the participants’ global knowledge in palliative care. Seventy-eight nurses (34.51%) showed excellent or good knowledge, and 148 (65.52%) poor or bad knowledge. The mode score was 8/20 (*n* = 46). Two hundred and twenty-six physicians (67.40%) showed excellent or good knowledge, with a response mode of 13/20 (*n* = 49), followed by 15/20 (*n* = 46). One hundred and nine physicians (32.53%) showed poor or bad knowledge.

No significant differences were observed according to palliative care training during undergraduate education in physicians. In contrast, nurses showed significant differences in the management of pain and respiratory disorders, and in global knowledge in palliative care (Table 3).

The professionals who carried out continuous training in palliative care in the last five years showed a significantly higher level of knowledge than those who did not, both globally and in its dimensions (Table 4). These significant differences were shown in physicians in all areas, except for psychiatric problems. In the case of nurses, significant differences (*p* < 0.05) were observed in the total score, and in the specific areas of philosophy and gastrointestinal problems assessed with the PCKT-SV.

In the same way, both in general and in its dimensions, professionals with experience in palliative care showed a significantly higher level of knowledge than those who did not have professional experience in palliative care (Table 5). This professional experience showed significant differences in four of the five dimensions evaluated in the PCKT-SV for physicians. In the case of nurses, previous experience showed significant differences with global knowledge and psychiatric problems.

A multiple regression model was performed to determine the influence of the independent variables (age, sex, profession, specialty, years of practice, undergraduate education, continuous training and work experience) on knowledge of palliative care in each of the participating healthcare professionals (physicians and nurses). The predictive factors included in the model that better explain the knowledge in palliative care of physicians (R2 = 17.7 F (7.321) = 11.06, *p* < 0.001) and nurses (R2 = 11.8 F (6.218) = 6.00, *p* < 0.001) are shown in Table 6 and Table 7, respectively.

According to this regression model, globally, younger people determined greater knowledge in the pain dimension in physicians (B(E) −0.09 (0.03) *t* = 2.82 *p* < 0.01). The years of professional practice of physicians significantly influenced global knowledge (B(E) 0.08 (0.03) *t* = 2.45 * *p* < 0.05). Continuous training (B(E) 1.68 (0.35) *t* = 4.73 *p* < 0.001) and work experience (B(E) 1.81 (0.41) *t* = 4.46 *p* < 0.001) were the best predictors to determine a significantly greater global knowledge of physicians in palliative care, also being significant in relation to the dimensions of philosophy, pain, respiratory disorders, and gastrointestinal disorders. Work experience also showed a significant influence on global knowledge and on the dimensions of pain, and respiratory and neuropsychiatric disorders (Table 6).

Regarding the nurses, age significantly influenced the knowledge of the dimensions of philosophy and neuropsychiatric disorders. Undergraduate education resulted in significantly greater overall knowledge (B(E) 1.65 (0.40) *t* = 4.09 *p* < 0.001), pain management (B(E) 1.01 (0.15) *t* = 6.55 *p* < 0.001), and respiratory disorder management (B(E) 0.53 (0.17) *t* = 3.17 *p* < 0.01). In this way, continuous training was a good predictor for philosophy of PCKT-SV (B(E) 0.35 (0.13) *t* = 2.75 *p* < 0.01), and work experience for neuropsychiatric disorders (B(E) 0.36 (0.15) *t* = 2.49 *p* < 0.05) (Table 7).

## 4. Discussion

The main objective of the present study was to assess the knowledge of healthcare professionals in palliative care using the PCKT-SV instrument. We also explored the differences between the health professionals involved. This study showed that physicians have a greater knowledge of palliative care than nurses, with no significant differences between sexes. Both continuous training and professional experience proved to be important values to determine significantly greater knowledge in palliative care in physicians and nurses.

There are different instruments that allow evaluation of the knowledge of professionals in the field of palliative care. Despite the fact that palliative care intervention is carried out in a multidisciplinary manner, in most cases, the existing evaluations focus on a single professional [27,28,29,30,31]. However, PCKT has been validated for both physicians and nurses [19]. Likewise, its psychometry is positive, with an interclass coefficient for the test–retest of 0.88 and an internal consistency of 0.81; its version adapted to Spanish used in the present study has similar results [23]. The PCKT has been cross-culturally adapted to different languages [21,22,23,24]; nevertheless, it has only been validated in Spanish [23] and an extended version in Germany [24].

In the present study, significant differences were observed between physicians and nurses (Table 2). The palliative care knowledge of physicians was significantly higher than that of nurses. However, the level of knowledge in the palliative care of nurses is increasing due to the inclusion of expert personnel in hospital teams [25]. The difference in knowledge between physicians and nurses is also related to the undergraduate training that they receive [32]. Furthermore, it must be taken into account that the duration of the undergraduate training is longer for physicians than for nurses. At any rate, previous studies have shown the need to increase training in this area in both professionals [16,17,21,33].

The participation of physicians in specific palliative care programs has shown a significant increase in this knowledge [34]. Likewise, physicians with a postgraduate degree, more than 10 years of experience, or who are more than 40 years old have significantly more knowledge of palliative care than their equals [21]. In the present study, the physicians and nurses who performed continuous training obtained significant differences in the global PCKT-SV score compared with the others (Table 4). Therefore, the existence of an ethical and deontological duty in the continuous training of health professionals for the benefit of our patients is required.

Several studies have analyzed the level of knowledge with the PCKT instrument. In this sense, the level of knowledge of physicians in Brazil proved to be below the expected average. Most of the participants (51.7%) obtained poor or bad levels of knowledge [22], with the dimension of philosophy being the best managed by these professionals. In contrast, in the present study, pain was the dimension with the best score in physicians. However, in Japan, the knowledge of physicians is significantly higher in palliative care, with a mean score of 72% [34]. Similar results were obtained valuing knowledge and attitudes in palliative care in Vietnam [21]. Physicians (35.1%) and nurses (74.2%) showed no adequate knowledge of palliative care in geriatrics. In the present study, low scores in pain, dyspnea, and gastrointestinal aspects were observed, especially in nurses, which was consistent with prior works [35,36]. The discrepancies observed in the PCKT results in different countries may be due to the cultural context or to the educational plans of each university.

Question 15 was the one that obtained a greater success rate. In contrast, question 6 (“The effect of opioids should decrease when pentazocine or buprenorphine hydrochloride is used together after opioids are used”) was the one with the highest error rate, in line with Ioshimoto et al. (2020) [22]. This could be justified by the lack of management of opioids such as pentazocine, since in Europe, treatment with other types of opioids is more usual.

In this study, a higher level of palliative care knowledge was found among physicians who were younger, with longer professional practice and palliative care work experience, and those who participated in continuous training. These results are in agreement with those of Mosich et al. (2017), where physicians with undergraduate palliative education and work experience were associated with higher scores in palliative knowledge [37]. Surprisingly, younger physicians showed greater knowledge. This could be due to the change in European educational programs in which physicians receive more training in palliative medicine during the degree. On the other hand, nurses’ undergraduate education had a significant weight on the level of knowledge (Table 7). In addition, nurses with continuous training in PC had better philosophical results, and nurses with work experience had a greater knowledge of neuropsychiatric disorders (Table 7). However, these results do not seem to occur in countries in which the discipline is novel [38].

Previous studies assessed both knowledge and attitude to death and dying in palliative care, emphasizing the relevance of teaching attitudes in palliative care [21,31,39]. In the present report, the attitude of the participants was not studied. However, a low correlation was shown between the attitudes of nursing professionals and their knowledge [17]. Other studies have reported that the attitude is a key element in the level of knowledge in palliative care [22,34], since a higher knowledge could be associated with a more positive attitude. Therefore, to improve the training of future professionals, it would be necessary to implement specific training in palliative care at all universities [32,40]. According to our results, nursing and medicine educational plans should be improved in order to cover specific palliative care knowledge gaps, including pain and dyspnea management for nurses, and philosophy dimension for physicians. In addition, curricular criteria for training should be consistent with those required by the European Association for Palliative Care (EAPC) to obtain the same training and skill level in the area [41]. Increasing training in palliative care could mean an improvement in the knowledge of physicians and nurses and in the management of professional skills and their attitudes.

The present study has some limitations. Our research was only conducted at the primary level of health services, further research is needed including different levels of care. The questionnaires were filled up online, the responses and profiles of the respondents could show differences comparing to questionnaires carried out face-to-face. In addition, the attitude of the participants has not been studied. To correlate attitudes with knowledge in palliative care, future research should analyze the attitudes of these professionals at the end of life.

## 5. Patents

Palliative Care Knowledge Test Spanish Version, PCKT-SV^®^ is registered in Safe Creative: https://www.safecreative.org/work/2010305757750-cuestionario-pckt-sv (access on 14 December 2020).

## 6. Conclusions

The results of this study showed that previous work experience, continuous training, and undergraduate training were essential factors in the acquisition of knowledge in palliative care. Physicians presented greater knowledge of palliative care than nurses, particularly in areas such as pain management, dyspnea, and gastrointestinal disorders. In contrast, nurses were more knowledgeable in the dimension of philosophy.

Knowledge of the palliative care of physicians and nurses presents deficiencies. The present study findings could have important curricular implications for physicians and nurses, as well practical implications in palliative care. Modifications in undergraduate and postgraduate training could increase the level of knowledge of nurses and physicians in palliative care, leading to better end-of-life patient care.

## Figures and Tables

**Figure 1 ijerph-18-05031-f001:**
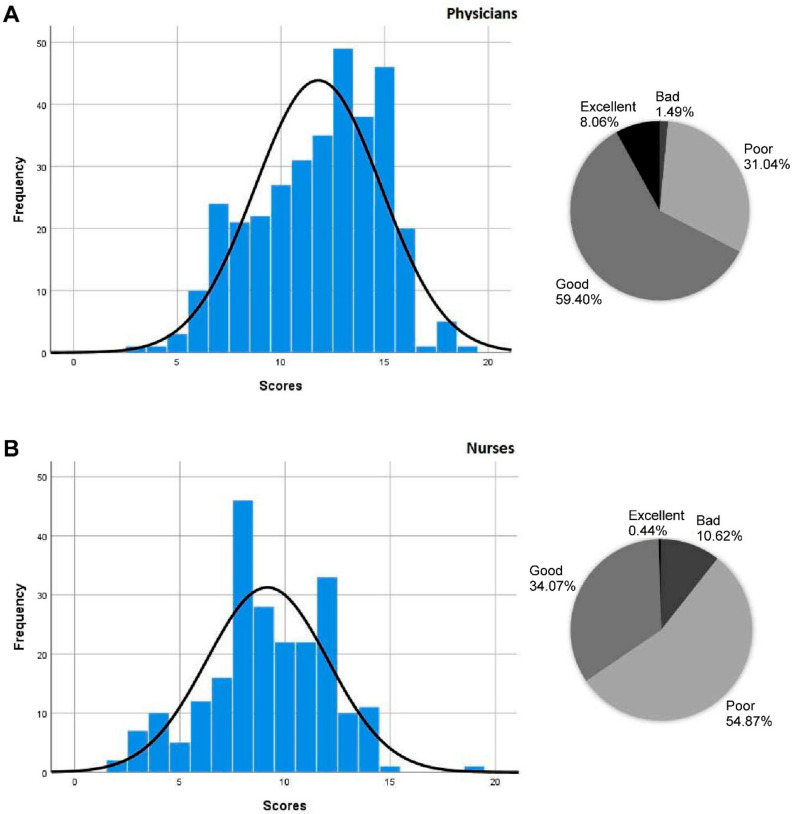
Distribution of knowledge based on PCKT-SV and healthcare professionals (**A**) Physicians; (**B**) Nurses. Percentage of correct answers: excellent, good, poor, and bad.

**Table 1 ijerph-18-05031-t001:** Characteristics of the sample.

Characteristics	Physicians		Nurses		Total		*p*-Value
	*n*	%	*n*	%	*n*	%	
Total	335	59.7	226	40.3	561	100	˂0.01
Sex							
Men	160	47.8	56	24.8	216	38.5	˂0.01
Women	175	52.2	170	75.2	345	61.5	
Undergraduate Palliative Training							
Yes	65	19.4	100	44.2	165	29.4	˂0.01
No	269	80.3	125	55.3	394	70.2	
DK/NA *	1	0.3	1	0.5	2	0.4	
Postgraduate/Continuous Training (last 5 years)							
Yes	136	40.6	40	17.7	176	31.4	˂0.01
No	198	59.1	186	82.3	384	68.4	
DK/NA *	1	0.3	0	0.0	1	0.2	
Work experience palliative care							
Yes	97	29	65	28.8	162	28.9	
No	235	70.1	161	71.2	396	70.6	
DK/NA *	3	0.9	0	0.0	3	0.5	
	Mean	SD	Mean	SD	Mean	SD	
Age	42.46	11.84	44.66	10.45	41.74	11.32	
Years of experience	14.85	10.90	15.76	8.83	15.22	10.12	

* DK/NA: Do not know/No answer Significance level: *p* < 0.05.

**Table 2 ijerph-18-05031-t002:** Score description of PCKT-SV for physicians and nurses.

Dimension	TOTAL Mean (SD)	Professional Average (SD*)*		Mean Difference	*t*-Test	
		Physicians (*n* = 335)	Nurses (*n* = 226)		*T* (559)	*p*-Value
Total (0–20)	10.7 (3.2)	11.85 (3.2)	9.16 (2.88)	2.69	10.16	<0.001
Philosophy (0–2)	1.3 (0.8)	1.14 (0.85)	1.47 (0.68)	−0.33	−4.86	<0.001
Pain (0–6)	2.5 (1.3)	2.95 (1.21)	1.93 (1.14)	1.02	10.02	<0.001
Dyspnea (0–4)	2.2 (1.2)	2.44 (1.16)	1.78 (1.15)	0.66	6.56	<0.001
Psychiatric disorders (0–4)	2 (0.9)	2.13 (0.85)	1.92 (0.92)	0.21	2.77	0.006
Gastrointestinal disorders (0–4)	1.9 (1.1)	1.9 (1.09)	1.92 (1.15)	−0.02	−0.22	0.82

Significance level: *p* < 0.05.

**Table 3 ijerph-18-05031-t003:** Scale scores of PCKT-SV according to palliative care training in the undergraduate education in physicians and nurses.

Undergraduate Education—Physicians					
Dimension	Mean (SD)		Mean Difference	*t*-Test	
	Yes (*n* = 65)	No (*n* = 269)		T (558)	*p*-Value
Total	12.05 (3.08)	11.74 (3.04)	0.30	0.71	0.47
Philosophy	1.30 (0.92)	1.10 (0.83)	0.20	1.72	0.08
Pain	2.78 (1.30)	2.98 (1.18)	−0.20	−1.20	0.23
Respiratory disorders	2.38 (1.28)	2.45 (1.13)	−0.06	−0.38	0.70
Neuropsychiatric disorders	2.15 (0.89)	2.13 (0.84)	0.02	0.20	0.84
Gastrointestinal disorders	1.68 (1.13)	1.96 (1.07)	−0.27	−1.85	0.64
**Undergraduate Education—Nurses**					
**Dimension**	**Mean (SD)**		**Mean difference**	***t*-Test**	
	**Yes (*n* = 100)**	**No (*n* = 125)**		**T (558)**	***p*-value**
Total	10.22 (2.71)	8.31 (2.75)	1.90	5.20	<0.001
Philosophy	1.49 (0.64)	1.46 (0.71)	0.26	0.28	0.777
Pain	2.48 (1.12)	1.48 (0.95)	1.00	7.21	<0.001
Respiratory disorders	2.14 (1.04)	1.50 (1.16)	0.64	4.32	<0.001
Neuropsychiatric disorders	1.97 (0.90)	1.88 (0.94)	0.09	0.72	0.468
Gastrointestinal disorders	1.89 (1.15)	1.94 (1.15)	0.04	−0.29	0.766

Significance level: *p* < 0.05.

**Table 4 ijerph-18-05031-t004:** Descriptive study and comparison of scores on PCKT-SV according to continuous training in palliative care in physicians and nurses.

Continuous Training—Physicians					
Dimension	Mean (SD)		Mean Difference	*t*-Test	
	Yes (*n* = 136)	No (*n* = 198)		*T* (558)	*p*-Value
Total	13.10 (2.62)	10.89 (3.0)	2.21	6.95	<0.001
Philosophy	1.31 (0.82)	1.03 (0.86)	0.27	2.91	0.004
Pain	3.26 (1.20)	2.73 (1.17)	0.53	4.04	<0.001
Respiratory disorders	2.77 (1.00)	2.20 (1.20)	0.56	4.50	<0.001
Neuropsychiatric disorders	2.23 (0.81)	2.07 (0.87)	0.35	1.80	0.072
Gastrointestinal disorders	2.11 (1.07)	1.76 (1.09)	0.35	2.93	0.004
**Continuous Training—Nurses**					
**Dimension**	**Mean (SD)**		**Mean Difference**	***t*-Test**	
	**Yes (*n* = 40)**	**No (*n* = 186)**		***T* (558)**	***p-*Value**
Total	10.3 (3.25)	8.97 (2.77)	1.05	2.10	0.036
Philosophy	1.75 (0.49)	1.41 (0.70)	0.34	2.88	0.004
Pain	2.15 (1.23)	1.88 (1.12)	0.27	1.35	0.179
Respiratory disorders	1.85 (1.23)	1.77 (1.14)	0.08	0.40	0.687
Neuropsychiatric disorders	1.95 (1.04)	1.91 (0.90)	0.03	0.19	0.849
Gastrointestinal disorders	2.25 (0.84)	1.85 (1.20)	0.40	2.01	0.045

Significance level: *p* < 0.05.

**Table 5 ijerph-18-05031-t005:** Descriptive study and comparison of PCKT-SV scores according to previous experience in palliative care in physicians and nurses.

Experience—Physicians					
Dimension	Mean (SD)		Mean difference	*t*-Test	
	Yes (*n* = 97)	No (*n* = 235)		*T* (558)	*p-*Value
Total	13.43 (2.43)	11.15 (3.03)	2.28	6.58	<0.001
Philosophy	1.24 (0.83)	1.11 (0.86)	0.13	1.22	0.221
Pain	3.40 (1.07)	2.77 (1.22)	0.64	4.47	<0.001
Respiratory disorders	2.85 (1.00)	2.26 (1.78)	0.59	4.29	<0.001
Neuropsychiatric disorders	2.37 (0.82)	2.03 (0.84)	0.34	3.34	0.001
Gastrointestinal disorders	2.15 (1.01)	1.80 (1.10)	0.35	2.66	0.008
**Experience—Nurses**					
**Dimension**	**Mean (SD)**		**Mean difference**	***t*-Test**	
	**Yes (*n* = 65)**	**No (*n* = 161)**		***T* (558)**	***p*-Value**
Total	9.95 (3.00)	8.84 (2.78)	1.115	2.66	0.008
Philosophy	1.52 (0.66)	1.45 (0.69)	0.07	0.69	0.488
Pain	2.03 (1.24)	1.89 (1.11)	0.143	0.84	0.398
Respiratory disorders	1.90 (1.29)	1.73 (1.09)	0.174	1.03	0.303
Neuropsychiatric disorders	2.14 (0.84)	1.84 (0.94)	0.30	2.23	0.027
Gastrointestinal disorders	2.15 (1.19)	1.83 (1.12)	0.327	1.95	0.052

Significance level: *p* < 0.05.

**Table 6 ijerph-18-05031-t006:** Multiple-dimensional regression models of the Palliative Care Knowledge Test-Spanish Version in physicians.

Variable	Total		Philosophy		Pain		Respiratory Disorders		Neuropsychiatric Disorders		Gastrointestinal Disorders	
	B(E)	*t*	B(E)	*t*	B(E)	*t*	B(E)	*t*	B(E)	*t*	B(E)	*t*
Age	−0.09 (0.03)	−2.82 **	0.02 (0.01)	1.60	0.03 (0.01)	2.16 *	−0.03 (0.01)	−1.87	−0.01 (0.01)	−0.71	0.00 (0.01)	−0.03
Gender (Male vs. Female)	0.09 (0.33)	0.27	0.01 (0.10)	0.05	0.06 (0.14)	0.42	0.23 (0.13)	1.76	−0.10 (0.10)	−0.98	−0.06 (0.13)	−0.45
Years of professional practice	0.08 (0.03)	2.45 *	−0.01 (0.01)	−1.03	−0.01 (0.01)	−0.52	0.01 (0.01)	1.06	0.01 (0.01)	0.58	0.02 (0.01)	1.16
Undergraduate Training (Yes vs. No)	−0.41 (0.40)	−1.00	0.27 (0.12)	2.22 *	−0.21 (0.17)	−1.24	−0.23 (0.16)	−1.39	−0.07 (0.12)	−0.56	−0.28 (0.15)	−1.82
Continuous Training (Yes vs. No)	1.68 (0.35)	4.73 ***	0.28 (0.11)	2.58 **	0.32 (0.15)	2.15 *	0.42 (0.14)	2.92 **	0.05 (0.11)	0.50	0.30 (0.14)	2.19 *
Work Experience (Yes vs. No)	1.81 (0.41)	4.46 ***	−0.14 (0.12)	−1.11	0.35 (0.17)	2.09 *	0.49 (0.16)	3.00 **	0.32 (0.12)	2.56 *	0.17 (0.16)	1.10
R^2^ (%)	17.7		3.1		10.4		9		2.5		4.8	
Model	F_7,321_ = 11.06 *p* < 0.001		F_7,321_ = 2.52 *p* = 0.015		F_7,321_ = 6.44 *p* < 0.001		F_7,321_ = 5.62 *p* < 0.001		F_7,321_ = 2.22 *p* = 0.032		F_7,321_ = 3.35 *p* = 0.002	

Significance level: * *p* < 0.05 ** *p* < 0.01 *** *p* < 0.001.

**Table 7 ijerph-18-05031-t007:** Multiple-dimensional regression models of the Palliative Care Knowledge Test-Spanish Version in nurses.

Variable	Total		Philosophy		Pain		Respiratory Disorders		Neuropsychiatric Disorders		Gastrointestinal Disorders	
	B(E)	*t*	B(E)	*t*	B(E)	*t*	B(E)	*t*	B(E)	*t*	B(E)	*t*
Age	0.03 (0.03)	1.03	0.02 (0.01)	2.00 *	−0.01 (0.01)	−0.87	0.01 (0.01)	0.39	0.02 (0.01)	2.44 *	0.00 (0.01)	−0.05
Gender (Male vs. Female)	−0.06 (0.46)	−0.13	−0.03 (0.11)	−0.22	−0.02 (0.18)	−0.14	0.08 (0.19)	0.44	0.06 (0.15)	0.39	0.02 (0.20)	0.08
Years of professional practice	−0.05 (0.04)	−1.49	−0.02 (0.01)	−1.96	0.01 (0.01)	0.63	−0.02 (0.01)	−1.36	−0.02 (0.01)	−1.80	−0.01 (0.02)	−0.59
Undergraduate Training (Yes vs. No)	1.65 (0.40)	4.09 ***	0.00 (0.10)	−0.04	1.01 (0.15)	6.55 ***	0.53 (0.17)	3.17 **	0.04 (0.14)	0.26	−0.15 (0.17)	−0.85
Continuous Training (Yes vs. No)	0.71 (0.51)	1.39	0.35 (0.13)	2.75 **	0.29 (0.20)	1.46	0.02 (0.21)	0.10	−0.14 (0.17)	−0.80	0.30 (0.22)	1.39
Work Experience (Yes vs. No)	0.69 (0.44)	1.59	−0.02 (0.11)	−0.21	−0.10 (0.17)	−0.62	0.10 (0.18)	0.58	0.36 (0.15)	2.49 *	0.27 (0.19)	1.47
R^2^ (%)	11.8		3		17.8		6.5		3.2		0.5	
Model	F_6,218_ = 6.00 *p* < 0.001		F_6,218_ = 2.15 *p* = 0.049		F_6,218_ = 9.08 *p* < 0.001		F_6,218_ = 3.58 *p* = 0.002		F_6,218_ = 2.22 *p* = 0.042		F_6,218_ = 1.19 *p* = 0.313	

Significance level: * *p* < 0.05 ** *p* < 0.01 *** *p* < 0.001.

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
