# Peer review of "Physicians’ and Nurses’ Knowledge in Palliative Care: Multidimensional Regression Models"

_ijerph, 2021, doi:10.3390/ijerph18095031_

Round 1
Reviewer 1 Report
The authors present a relevant issue. However, some important data is missing in the description of the sample and the analysis of results that should be included for publication.
In general, the manuscript is well written.
Here are some suggestions:
ABSTRACT
In the abstract, delete "and 561 were 21 answered". This data is repeated in the next sentence with the description of the sample.
INTRODUCTION
In the introduction the authors have linked the aging of the population with the need for palliative care. Although this aging will lead to more comorbidities and complex patients, palliative care is not an exclusive need of the aging population.This should be clear in this section.
RESULTS
In this section the authors stated that of the 600 questionnaires distributed, 588 questionnaires were answered and 39 rejected. I'm not sure about what they mean. With rejected , Do they refer to people who refused to participate in the study?. If so, I do not understand that 588 questionnaires were answered. Was there any missing data on any of those answered questionnaires?
Before carrying out a comparison of the test results between both groups (physicians and nurses), it does not seem to have been compared whether the samples were similar before answering the questionnaire, in terms of age, experience, training, etc. Since that (and not the type of degree) could explain the results in relation to the level of knowledge.
It would be important to introduce a table with the characteristics of the sample, including years of work experience, training received in palliative care in the degree, continuous training received in the last 5 years and work experience in palliative care (since many of these data do not appear later in the results). In addition, the large standard deviations of some data must be taken into account. I also do not understand that 73-year-olds have answered the questionnaire (I understand that they are no longer practicing and I do not know the relevance of this for clinical practice).
On page 4 there appears to be a misprint on line 143. "A total of 65.52% (n = 148) of professional nurses showed 143 poor or poor knowledge". I guess they meant bad or poor. Anyway, when comparing the results, if one is talking about the poor level of knowledge of one group (in this case nursing), it should be compared with the poor level of the other, and not present the bad data of one group by comparing them with the good ones of the other.
Furthermore, taking into account the duration of the training of physicians, it is understood (not clear at all in the manuscript) that the youngest participants in this study (and therefore with less training and experience) are nurses. In short, a separate analysis by groups does not seem to make sense if the results of the variables used for the analysis in each group are not previously presented.
I am not very clear what table 2 presentes (the method should better describe how this data is extracted). I suppose they have asked the participants if they had specific training at the university, on the dimensions measured by the questionnaire.
DISCUSSION
The discussion of the study should be considerably improved. The authors present data such as the mean experience necessary to improve knowledge (which has not been shown by groups in this study), or the differences in the results according to the country where the study is carried out, without, for example, in this case, look for a possible cultural explanation
The authors' explanation for the success rate in the response to item 6 seems surprising since, in theory, they have previously carried out an adaptation and cultural validation of the scale with the Spanish population, so it is understood that this item should have already been corrected and the instrument piloted before sample collection.
The results of the regression model should appear in that section, not in the discussion. They could also present a graph.
Other data are presented that have not been sufficiently discussed. For example, in relation to the specific training of undergraduate nurses, a lack of consensus has been demonstrated in the study plans at the national level, with many universities without a study plan that considers this training as a specific subject. Then they refer to the possible influence of other factors not measured in this study, such as attitude, but it is not explained what is understood by attitude in this sense or what is the specific influence of this variable.
Taking into account also that in the case of nurses, continuous training does not seem to have an effect in certain dimensions, in the discussion they should have proposed the need to review the educational plans or the contents of the courses offered in this regard, which consider covering specific gaps in knowledge related to palliative care.
If the questionnaire has been answered online, then it should also be considered a limitation since previous studies have shown differences in the responses and profile of the respondent when the studies are carried out online.
Author Response
Reviewer 1
Comment: The authors present a relevant issue. However, some important data is missing in the description of the sample and the analysis of results that should be included for publication.
In general, the manuscript is well written.
Response: We are very grateful for this positive comment. All your comments have been included in the revised manuscript and explained point-by-point below.
Here are some suggestions:
Comment 1:
ABSTRACT
In the abstract, delete "and 561 were 21 answered". This data is repeated in the next sentence with the description of the sample.
Response:
The abstract has been changed according to your suggestion (lines 21-22).
INTRODUCTION
Comment 2: In the introduction, the authors have linked the aging of the population with the need for palliative care. Although this aging will lead to more comorbidities and complex patients, palliative care is not an exclusive need of the aging population. This should be clear in this section.
Response: We agree with the reviewer's comment. The palliative cares not only are associated with the aging population. Regardless of the aging, palliative cares are also needed in other diseases like chronic diseases, cancer, heart and respiratory diseases among others. This statement has been clarified in the Introduction section including a new reference (lines 39 to 40).
RESULTS
Comment 3: In this section, the authors stated that of the 600 questionnaires distributed, 588 questionnaires were answered and 39 rejected. I'm not sure about what they mean. With rejected, Do they refer to people who refused to participate in the study?. If so, I do not understand that 588 questionnaires were answered. Was there any missing data on any of those answered questionnaires?
Response: Thank you for this suggestion, we have clarified it in the revised manuscript (line 77-79, Material and Methods sections, Setting). In our study, a total of 600 PCKT-SV survey questionnaires were distributed among health professionals (physicians and nurses), 588 of them sent the questionnaires back. After a revision, we verified that 561 of them were properly filled.
Comment 4: Before carrying out a comparison of the test results between both groups (physicians and nurses), it does not seem to have been compared whether the samples were similar before answering the questionnaire, in terms of age, experience, training, etc. Since that (and not the type of degree) could explain the results in relation to the level of knowledge.
It would be important to introduce a table with the characteristics of the sample, including years of work experience, training received in palliative care in the degree, continuous training received in the last 5 years and work experience in palliative care (since many of these data do not appear later in the results). In addition, the large standard deviations of some data must be taken into account. I also do not understand that 73-year-olds have answered the questionnaire (I understand that they are no longer practicing and I do not know the relevance of this for clinical practice).
Response: Thank you for this suggestion. We have included a Table (Table 1) in the revised manuscript (Material Section, Instrument) showing characteristics of the participants (Sex, Age, Work experience, undergraduate training in palliative care, continuous training and work experience in palliative care). (Line 97)
Regarding the participant with 73 years old, is an emeritus health professional practicing during the study, so his participation was considered pertinent.
Comment 5: On page 4 there appears to be a misprint on line 143. "A total of 65.52% (n = 148) of professional nurses showed 143 poor or poor knowledge". I guess they meant bad or poor. Anyway, when comparing the results, if one is talking about the poor level of knowledge of one group (in this case nursing), it should be compared with the poor level of the other, and not present the bad data of one group by comparing them with the good ones of the other.
Furthermore, taking into account the duration of the training of physicians, it is understood (not clear at all in the manuscript) that the youngest participants in this study (and therefore with less training and experience) are nurses. In short, a separate analysis by groups does not seem to make sense if the results of the variables used for the analysis in each group are not previously presented.
Response: We have corrected the misprint, and we have used the Ioshimoto et al. (2020) score in the whole revised text (line 160 in the revised manuscript). Regarding to the second comment, we really appreciate your statement. In this sense, we have changed the text in the results section, presenting the same level of knowledge for both group of professionals (nurses and physicians) (lines 159-163). Concerning to the third comment, we have included the variables used for the analysis in the revised manuscript (Table 1). In the table, it’s showed that the mean age of physicians (42.46 ± 11.84) and nurses (44.66 ± 10.45) are very similar.
Comment 6: I am not very clear what table 2 presents (the method should better describe how this data is extracted). I suppose they have asked the participants if they had specific training at the university, on the dimensions measured by the questionnaire.
Response: Table 2 (now Table 3) presents participants scores of PCKT-SV according to palliative care training in the undergraduate education in physicians and nurses. We have clarified the data collection for this study and table in the Methods Section as follows (lines 105 to 109):
“Undergraduate training was considered when the participants stated to have received theoretical or practical training on the dimensions measured in the questionnaire during their university studies. Postgraduate and continuous training was considered when the health professionals stated to have participated in courses/masters on palliative care after the degree.”
DISCUSSION
Comment 7: The discussion of the study should be considerably improved. The authors present data such as the mean experience necessary to improve knowledge (which has not been shown by groups in this study), or the differences in the results according to the country where the study is carried out, without, for example, in this case, look for a possible cultural explanation.
Response: The discussion has been deeply modified according to all your suggestions. Regarding the experience necessary to improve knowledge, we have clarified in the discussion section that continuous training and professional experience proved to be important values to determine greater knowledge in palliative care in both groups, physicians and nurses (Lines 232-233).
Concerning the cultural differences in the results of the PCKT in different countries, we have included the following sentence in the Discussion section: “However, all these discrepancies in the PCKT results in different countries may be due to cultural contexts or to the various educational plans in universities, among others.” (Lines 269- 271).
Comment 8: The authors' explanation for the success rate in the response to item 6 seems surprising since, in theory, they have previously carried out an adaptation and cultural validation of the scale with the Spanish population, so it is understood that this item should have already been corrected and the instrument piloted before sample collection.
Response: We totally agree with the Reviewer 1, however the Spanish version of the PCKT used in this study had been already validated in our cultural context in a previous study. To properly change the item 6, a validation study will be necessary done in future revisions.
Comment 9: The results of the regression model should appear in that section, not in the discussion. They could also present a graph.
Response: The results of the regression models have been moved to the results section (lines 198-201). We have checked the possibility of present the regression model as a graph; however, it has to be combined 5 dimensions with 226 nurses and 335 physicians. This combination would be difficult to understand in the graph, being the results clearer presented as a table.
Comment 10: Other data are presented that have not been sufficiently discussed. For example, in relation to the specific training of undergraduate nurses, a lack of consensus has been demonstrated in the study plans at the national level, with many universities without a study plan that considers this training as a specific subject. Then they refer to the possible influence of other factors not measured in this study, such as attitude, but it is not explained what is understood by attitude in this sense or what is the specific influence of this variable.
Response: We have explained, discussed and included the meaning of attitude and the influence of this variable on knowledge in palliative care (lines 291-292 and 296). These statements have been supported by the references:
- Thi Thanh Vu, H.; Hoang Nguyen, L.; Xuan Nguyen, T.; Hoai Nguyen, T.T.; Ngoc Nguyen, T.; Thu Nguyen, H.T.; Trung Nguyen, A.; Pham, T.; Tat Nguyen, C.; Xuan Tran, B.; Latkin, C.A-; Ho, C.S.H.; Ho, R.C.M. Knowledge and Attitude Toward Geriatric Palliative Care among Health Professionals in Vietnam. Int J Environ Res Public Health 2019, 16, 2656. DOI: 10.3390/ijerph16152656
- Cevik B, Kav S. Attitudes and experiences of nurses toward death and caring for dying patients in Turkey. Cancer Nurs. 2013 Nov-Dec;36(6):E58-65. doi: 10.1097/NCC.0b013e318276924c. PMID: 23151504
- Al-Ansari, A.M.; Suroor, S.N.; AboSerea, S.M.; Abd-El-Gawad, W.M. Development of palliative care attitude and knowledge (PCAK) questionnaire for physicians in Kuwait. BMC Palliat Care 2019, 18, 49. DOI: 10.1186/s12904-019-0430-9
Comment 11: Taking into account also that in the case of nurses, continuous training does not seem to have an effect in certain dimensions, in the discussion they should have proposed the need to review the educational plans or the contents of the courses offered in this regard, which consider covering specific gaps in knowledge related to palliative care.
Response: As Reviewer 1 suggested, we have literally introduced the following sentence in the Discussion section: “In the case of nurses, according to our results, there is a need to review the educational plans or the contents of the courses offered in pain and dyspnea management, to cover specific knowledge gaps related to palliative care.” (Lines 298-303)
Comment 12: If the questionnaire has been answered online, then it should also be considered a limitation since previous studies have shown differences in the responses and profile of the respondent when the studies are carried out online.
Response: The limitation has been included in the Discussion section of the revised manuscript (Lines 309-312).
Reviewer 2 Report
Dear authors.
Thank you for the opportunity to review your article.
I consider the topic relevant and actual.
Despite worldwide recommendations to strengthen the pre and postgraduate training of health professionals on palliative care, the results of your study contribute to deepen the knowledge about the effectiveness of this advice.
After reading your document, my suggestions for improvement are:
- Page 5: “No significant differences were observed according to palliative care training during undergraduate education in physicians. In contrast, nurses showed significant differences in the management of pain and respiratory disorders and in global knowledge in palliative care”. I suggest you clarify in methodology what did you consider “palliative care training during undergraduate education”, specify how you measured the existence of pre-graduate training versus no training (e.g. clinical training, academic contents).
- References number 1 and 2 should have the publication data, namely WHO (2018) and WHO (2014)
I believe that this article can be published with minimal corrections.
Author Response
REVIEWER 2
Open Review
Dear authors.
Comment 1: Thank you for the opportunity to review your article.
I consider the topic relevant and actual.
Despite worldwide recommendations to strengthen the pre and postgraduate training of health professionals on palliative care, the results of your study contribute to deepen the knowledge about the effectiveness of this advice.
Response: We thanks the Reviewer 2 for his/her kind comment.
After reading your document, my suggestions for improvement are:
- Comment 1: Page 5: “No significant differences were observed according to palliative care training during undergraduate education in physicians. In contrast, nurses showed significant differences in the management of pain and respiratory disorders and in global knowledge in palliative care”.I suggest you clarify in methodology what did you consider “palliative care training during undergraduate education”, specify how you measured the existence of pre-graduate training versus no training (e.g. clinical training, academic contents).
Response: Undergraduate training was considered when the participants stated to have received theoretical or practical training on the dimensions measured by the questionnaire during their university studies. Postgraduate and continuous training was considered when the health professionals stated to have participated in courses on palliative care after the degree This statement has been clarified in the Methods section (lines 105-109).
- Comment 2: References number 1 and 2 should have the publication data, namely WHO (2018) and WHO (2014)
Response: We have modified the references 1 and 2 as the Reviewer 2 suggested.
I believe that this article can be published with minimal corrections.
Response: Thank you for your kind comment.
Round 2
Reviewer 1 Report
The authors have reviewed each comment and made the pertinent modifications. Overall, the manuscript appears to be well assembled.
There are still some small aspects that should be reviewed before publication.
In response to possible differences between groups according to sociodemographic characteristics, the authors have included a table (Table 1) in the results, however no analysis has been included comparing possible significant differences between the groups.
In section 2.4 the authors explain what each variable means but I cannot find an explanation for the experience. Was a minimum time of experience considered so that the participant could indicate whether or not they had experience (for example, 6 months working in palliative care)?
I still do not understand, if the scale has already been reviewed and validated in our population, why do they consider that it has to be validated in the future? Didn't they take into account cultural differences in their adaptation?
Finally, it would be recommended that the modifications were proof-read by a native speaker.
Author Response
COMMENTS TO REVIEWER 1
We appreciate the reviewer`s comments. Addressing them we improve the paper.
Comment 1: The authors have reviewed each comment and made the pertinent modifications. Overall, the manuscript appears to be well assembled.
Response: We thank the reviewer 1 for his/her kind comment.
There are still some small aspects that should be reviewed before publication.
Comment 2: In response to possible differences between groups according to sociodemographic characteristics, the authors have included a table (Table 1) in the results, however no analysis has been included comparing possible significant differences between the groups.
Response: Table 1 has been moved to the Results section (line 149) and we have included significant differences between groups. We have also included some sentences that clarify the data presented in Table 1 in the revised manuscript (lines 134-138).
Comment 3: In section 2.4 the authors explain what each variable means but I cannot find an explanation for the experience. Was a minimum time of experience considered so that the participant could indicate whether or not they had experience (for example, 6 months working in palliative care)?
Response: Our research was conducted at the primary level of health services and the experience in palliative care is included in their general clinical assistance, therefore there isn’t a minimum time of palliative care experience. The variable of work experience in palliative care was defined as the assistance to any patient with palliative care requirement. The definition of this variable has been included in section 2.4 (lines 105-106)
Comment 4: I still do not understand, if the scale has already been reviewed and validated in our population, why do they consider that it has to be validated in the future? Didn't they take into account cultural differences in their adaptation?
Response: Thank you for this precise comment. The scale has been validated in our population and cultural differences have been taken into account. We have eliminated that consideration in the revised manuscript.
Comment 5: Finally, it would be recommended that the modifications were proof-read by a native speaker
Response: The modifications have been proof-read by a native speaker. Changes are highlighted in yellow.
